High matrix metalloproteinase-2 expression predicts poor prognosis of colon adenocarcinoma and is associated with PD-L1 expression and lymphocyte infiltration

Xiao Yiyi 1 2
Li Guangming 3
Xie Yongjie 4
Shao Bo 1 2
Hao Jingpeng 5
Zhu Yanglin 1 2 6
Kong Dejun 7
Qin Yafei 1 2 8
Qin Hong 1 2 9
Ren Shaohua 1 2 10
Wang Hongda 1 2
Sun Chenglu 1 2
Wang Hao 1 2 11 hwangca272@hotmail.com
1 Department of General Surgery, Tianjin Medical University General Hospital , Tianjin , China
2 Tianjin General Surgery Institute, Tianjin Medical University General Hospital , Tianjin , China
3 Department of General Surgery, The Second Affiliated Hospital of Guangzhou Medical University , Guangzhou , China
4 Department of Pancreatic Oncology, Tianjin Medical University Cancer Institute and Hospital , Tianjin , China
5 Department of Colorectal Surgery, The Second Hospital of Tianjin Medical University , Tianjin , China
6 Department of Hepatobiliary Pancreatic Gastrointestinal Surgery, Jinhua Hospital of Wenzhou Medical University , Jinhua , China
7 School of Medicine, Nankai University , Tianjin , China
8 Department of Vascular Surgery, Henan Provincial People’s Hospital , Zhengzhou , China
9 Department of Breast and Thyroid Surgery, the First College of Clinical Medical Science, China Three Gorges University , Yichang , China
10 Department of General Surgery, The Affiliated Hospital of Inner Mongolia Medical University , Hohhot , China
11 Tianjin Key Laboratory of Precise Vascular Reconstruction and Organ Function Repair , Tianjin , China
Uversky Vladimir
Electronic publication date: 2025 Jun 30
Publication date: 2025
Volume: 13
Electronic Location ID: e19550
Received 2024 Nov 5; Accepted 2025 May 12
Copyright: © 2025 Xiao et al.
Copyright year: 2025
Copyright holder: Xiao et al.
License: This is an open access article distributed under the terms of the Creative Commons Attribution License, which permits unrestricted use, distribution, reproduction and adaptation in any medium and for any purpose provided that it is properly attributed. For attribution, the original author(s), title, publication source (PeerJ) and either DOI or URL of the article must be cited.
License URL: https://creativecommons.org/licenses/by/4.0/

Keywords: Colon adenocarcinoma, Immunotherapy, Matrix metalloproteinase-2, PD-L1, Lymphocyte infiltration

Funding: National Natural Science Foundation of China 82071802, 82270794 Natural Science Foundation of Tianjin 21JCYBJC00850 Science and Technology Project of Tianjin Health Commission TJWJ2021MS004 Tianjin Key Medical Discipline (Specialty) Construction Project TJYXZDXK-076C This work was supported by grants to Hao Wang from the National Natural Science Foundation of China (No. 82071802, and 82270794), Natural Science Foundation of Tianjin (No. 21JCYBJC00850), Science and Technology Project of Tianjin Health Commission (No. TJWJ2021MS004), and Tianjin Key Medical Discipline (Specialty) Construction Project (TJYXZDXK-076C). The funders had no role in study design, data collection and analysis, decision to publish, or preparation of the manuscript.

==============================
Background

Colon adenocarcinoma (COAD) is a prevalent and aggressive malignancy with limited treatment options, particularly for advanced stages. While programmed death-ligand 1 (PD-L1) inhibition, has emerged as an appealing therapeutic approach for COAD, its effectiveness as a monotherapy is hindered by high tumor heterogeneity. Identifying novel therapeutic targets to boost the efficacy of PD-L1-based immunotherapy in COAD is crucial to improving clinical outcomes. Matrix metalloproteinase-2 (MMP-2), traditionally known for its role in tumor invasion, metastasis, and angiogenesis, has not been thoroughly investigated in the relationship to immunotherapy for COAD. This work aims to investigate the potential involvement of MMP-2 in the immune microenvironment of COAD and explore its possible role as a target to enhance the therapeutic efficacy of anti-PD-L1-based immunotherapy.

Methods

This study employed a comprehensive bioinformatics analysis of publicly available datasets to investigate the correlation between MMP-2 expression and PD-L1 levels in COAD. Additionally, we evaluated the impact of MMP-2 expression on patient survival and prognosis. To validate these findings, in vitro experiments were conducted to assess the effect of MMP-2 inhibition on PD-L1 expression in colon cancer cell lines. We also analyzed the association between MMP-2 expression and tumor-infiltrating lymphocytes (TILs) to elucidate the immunological landscape of COAD.

Results

Our bioinformatic analysis revealed a novel positive correlation between MMP-2 expression and PD-L1 level in COAD, indicating that higher MMP-2 level is associated with increased PD-L1 expression. Furthermore, in COAD patients, elevated MMP-2 expression was linked to poor overall survival and prognosis. In vitro experiments demonstrated that inhibiting MMP-2 significantly reduced PD-L1 expression in SW480 cells, suggesting that MMP-2 plays a regulatory function in immune evasion. In addition, a novel negative relationship between MMP-2 expression and the presence of TILs was identified, underscoring MMP-2’s potential role in modifying the COAD immunological landscape.

Conclusion

This work shows for the first time that MMP-2 not only contributes to tumor progression but also plays a critical role in the immunosuppressive microenvironment of COAD. The demonstrated association between MMP-2 and PD-L1 expression, along with its effect on TILs, indicates that MMP-2 is a promising alternative target for improving the efficacy of anti-PD-L1 immunotherapy. Targeting MMP-2 may offer a novel avenue for overcoming resistance to conventional immunotherapies, potentially improving treatment outcomes in COAD patients.

Introduction

Colorectal cancer (CRC) is the third most commonly diagnosed malignant tumor worldwide, contributing to almost 10% of all cancer-related fatalities (Arnold et al., 2017; Kuipers et al., 2015). Its burden is anticipated to rise dramatically, with projections estimating a 60% increase in the incidence of CRC by 2030, reaching over 2.2 million new cases and 1.1 million deaths (Arnold et al., 2017). Colon adenocarcinoma (COAD), the predominant histological subtype of colon cancer, has a significantly higher incidence and mortality risk compared to rectal cancer (Bray et al., 2018). These worrisome statistics highlight the pressing requirement for more potent therapeutic strategies to improve outcomes in COAD.

Immunotherapy is thought to be a powerful and promising therapeutic approach in multiple malignancies, such as melanoma, non-small cell lung cancer and COAD (Passardi et al., 2017; Reck et al., 2016; Robert et al., 2015). Specially, among the various immunotherapeutic strategies, immune checkpoint inhibitors (ICIs) play a critical role in the treatment of cancers (Das & Johnson, 2019). The inhibitors targeting programmed death 1 (PD-1) and its ligand programmed death ligand 1 (PD-L1), such as nivolumab (Brahmer et al., 2010; Yamamoto et al., 2017), pembrolizumab (Patnaik et al., 2015) or MPDL3280A (Herbst et al., 2014), have shown the prospect of improving COAD clinical outcomes by reactivating anti-tumor immune responses. Nevertheless, objective response rates to anti-PD-L1 therapy remain limited in solid tumors, including COAD; however, the underlying mechanisms remain unclear (Brahmer et al., 2012; Das & Johnson, 2019). This limited response is attributed to the highly immunosuppressive tumor microenvironment (TME) and the complex molecular heterogeneity of COAD, which affects both tumor immunogenicity and immune cell infiltration. Recent studies have demonstrated that combination treatments can boost the efficacy of monotherapies in overcoming resistance to ICIs. For example, the combination of nivolumab (anti-PD-1) and ipilimumab (anti-cytotoxic T-lymphocyte-associated protein 4, anti-CTLA-4) significantly improves overall survival (OS) and progression-free survival in metastatic CRC patients compared to the treatment with nivolumab alone (Overman et al., 2016). Mitogen-activated protein kinase (MEK) inhibition, on the other hand, has been shown to encourage MHC-I expression and promote T cell infiltration, both of which enhance the efficacy of anti-PD-L1 therapy (Bendell et al., 2016). Additionally, Ubiquitin-specific peptidase 8 (USP8) inhibition markedly boosted and activated the CD8+ T cells through increasing the expression of PD-L1 in tumor cells and triggering the NF-κB pathway, thereby inhibiting tumor growth, improving the outcomes of anti-PD-L1 therapy (Xiong et al., 2022). Although researchers have identified numerous targeted medications aimed to enhancing the efficacy of anti-PD-L1 therapy, the therapeutic response in COAD remains inconsistent (Topalian et al., 2016). Consequently, there is an imperative need to identify and validate novel molecular targets that can optimize and potentiate the efficacy of anti-PD-L1 immunotherapy in COAD patients.

Matrix metalloproteinases (MMPs), a family of 24 proteolytic enzymes, play critical roles in tumor invasion, metastasis, and neoangiogenesis by degrading various components of the extracellular matrix (ECM) (Winer, Adams & Mignatti, 2018). MMP-2, in particular, has been implicated in lymphatic invasion, and lymph node metastasis (Langenskiold et al., 2005; Quintero-Fabian et al., 2019). A research has shown that the inhibition of MMP-2 and MMP-9 can diminish angiogenesis and lymphangiogenesis, as well as suppress lymph node metastasis (Nakamura et al., 2004). Additionally, MMPs have been demonstrated to affect the immune microenvironment in tumors by influencing immune cell infiltration and function (Kessenbrock, Plaks & Werb, 2010; Li et al., 2016). For the last few decades, MMP inhibitors (MMPIs) like batimastat, marimastat, and tanomastat have been applied into cancer clinical trials for various cancer types (Winer, Adams & Mignatti, 2018). Unfortunately, these agents have failed to improve the OS of cancer patients while causing severe side such as joint pain, pyrexia, dyspnea, and cough (Winer, Adams & Mignatti, 2018). The unsatisfactory results may be attributed to the drugs’ non-specificity and the complexity of cancer biology. Some medications even block several MMPs with anti-tumorigenic activity, including MMP-3, MMP-9, and MMP-11 (Dufour & Overall, 2013). Thus, understanding the specific functions of MMPs, especially MMP-2, in COAD may pave the way for more targeted therapies.

CD8+ T cells, a critical component of the anti-tumor immune response, also known as cytotoxic T lymphocytes, are pivotal in recognizing and eliminating tumor cells (Sharma, Rive & Holt, 2019). In CRC, numerous studies have been conducted on the presence and activity of CD8+ T lymphocytes in the tumor microenvironment, and they are strongly associated with patient prognosis and clinical outcomes. Their infiltration within the tumor microenvironment has been inversely correlated with tumor progression, and low CD8+ T cell infiltration is associated with poorer clinical outcomes in COAD patients, emphasizing their importance in mediating anti-tumor immunity (Galon et al., 2006; Hartman et al., 2020; Li et al., 2020). Their infiltration is a predictive marker for positive responses to ICIs like pembrolizumab and nivolumab (Overman et al., 2017). Nonetheless, factors like the ECM and tumor-derived molecules such as transforming growth factor beta (TGF-β) can inhibit CD8+ T cell activity, contributing to an immunosuppressive microenvironment (Joyce & Fearon, 2015; Mariathasan et al., 2018).

In this study, our emerging evidence suggests that high expression of MMP-2 in COAD is correlated with poor prognosis and reduced survival, likely due to its role in modulating the immune microenvironment by influencing immune cell infiltration and PD-L1 expression. Despite these findings, the potential of MMP-2 as a therapeutic target in COAD remains underexplored. This study aims to elucidate the impact of MMP-2 on COAD progression and its interplay with immune checkpoint pathways. Understanding the contribution of MMP-2 to tumor immune evasion and its association with poor clinical outcomes could provide a compelling rationale for targeting MMP-2 in combination with ICIs, potentially offering a novel therapeutic strategy to overcome resistance and improve the efficacy of immunotherapy in COAD.

Materials and Methods

Microarray data collection and preprocessing

Transcriptome data and corresponding clinical information of COAD patients were sourced from the Gene Expression Omnibus (GEO) database. Two datasets, GSE197802 (33 COAD samples, processed on GPL18573 Illumina NextSeq 500, Homo sapiens) and GSE140973 (12 COAD samples, processed on GPL17077 Agilent-039494 SurePrint G3 Human GE v2 8 × 60 K Microarray 039381), were selected for analysis. After background correction and data normalization, the gene expression matrices of the two datasets were obtained. Additionally, transcriptomic data from 440 COAD samples in The Cancer Genome Atlas (TCGA; https://www.cancer.gov/ccg/research/genome-sequencing/tcga) database were included for further analysis. Samples lacking clinical data were excluded.

Evaluation of immune cell infiltration

To estimate the composition and abundance of immune cells within the tumor microenvironment, we employed the CIBERSORT algorithm (Newman et al., 2015; https://cibersortx.stanford.edu/). CIBERSORT deconvolves the transcriptomic expression matrix using linear support vector regression, providing an estimate of immune cell populations. Gene expression matrix data were uploaded to the CIBERSORT web portal, and samples with a p-value < 0.05 were retained for downstream analysis. The distribution of 22 immune cell types across samples was visualized with histogram generated by the “ggplot2” package. Moreover, Gene Set Variation Analysis (GSVA) was performed using the immune-related gene set “C7.all.v7.2.symbols.gmt” to assess immune activity, with significance enrichment defined by p < 0.05 and false discovery rate (FDR) < 0.25.

Differentially expressed gene screening

Differentially expressed genes (DEGs) from GSE197802, GSE140973, and TCGA datasets were screened by the “limma” package (cutoff |Log2FC| > 1, p < 0.05). The results were visualized using both Volcano plot and heatmaps. The volcano map was generated by the “ggplot2” package and the heatmap was obtained by the “heatmap” package. Then the genes associated with low CD8+ T cell infiltration were further identified by intersecting the above up-regulated DEGs.

Functional analysis

Functional annotation of the DEGs was performed using gene ontology (GO) and Kyoto Encyclopedia of Genes and Genomes (KEGG) pathway enrichment analysis through the “clusterProfiler” package in R. GO enrichment analysis covered cellular component (CC), molecular function (MF), and biological process (BP). Statistically significant enrichment was defined as unadjusted p < 0.01, minimum count > 3, enrichment factor > 1.5, and adjusted p < 0.05. The Gene Set Enrichment Analysis (GSEA; https://www.gsea-msigdb.org/gsea/index.jsp) was used to enrich differentially expressed mRNA pathways, and 10,000 permutations were performed for each analysis. The KEGG Pathways dataset from curated gene sets was selected for this purpose. The threshold for the statistically significant GSEA analysis was set to the corrected p < 0.05 and FDR < 0.25. The reference gene set used was “c2.cp.kegg.v7.0.symbols.gmt”. p < 0.05 and FDR < 0.25 were considered to be significantly enriched. The result of enrichment analysis would be characterized by corrected p values and normalized enrichment scores (NES). GSEA enrichment analysis and visualization were performed using GSEA local software. To mitigate potential biases in GSEA due to sample grouping, we employed stringent quality control measures, including the normalization of gene expression data and the verification of sample distribution prior to analysis. These steps ensured the reliability of our comparative pathway enrichment findings.

Cell culture and transfection

The human colon cancer cell line SW480 (Procell Life Science & Technology, Wuhan, China, batch of cells is CL-0223) and Caco-2 (Procell Life Science & Technology, Wuhan, China, batch of cells is CL-0050) were cultured in RPMI-1640 medium (HyClone, Logan, UT, USA), supplemented with 10% fetal bovine serum (FBS), 100 U/mL of penicillin, and 100 μg/mL streptomycin. SW480 cells and Caco-2 cells were seeded in 6-well plates for transfection. The adenoviral constructs designed for this study included siRNA targeting MMP-2 (referred to as MMP2-siRNA) and a scrambled sequence for MMP-2 (referred to as Control siRNA). These constructs were created as previously described (Chetty et al., 2006). The specific sequence used for the MMP2-siRNA is 5′-AACGGACAAAGAGTTGGCAGTATCGATACTGCCAACTCTTTGTCCGTT. While the scrambled sequence used for the Control siRNA is 5′-GCACGGAGGTTGCAAAGAATAATCGATTATTCTTTGCAACCTCCGTGC. The specific siRNA targeting MMP-2 and the control siRNA (Tsingke Biotechnology Co., Ltd., Beijing, China) was prepared by mixing 100 pmol of siRNA with 120 µL serum-free RPMI-1640 medium. Separately, 5 µL Lipofectamine 2000 (Invitrogen, Carlsbad, California, USA) was combined with 120 µL serum-free RPMI-1640 medium and incubated for 5 min at room temperature. The above siRNA and Lipofectamine 2000 mixtures were then combined and incubated for 20 min at room temperature before being added to the cells. Subsequently, the cells were cultured at 37 °C by adding 750 uL of serum-free RPMI-1640 medium to the mixtures. After 6 h of transfection, the medium was replaced with RPMI-1640 supplemented with 10% FBS. Transfected cells were harvested 48 h post-transfection for subsequent experiments. Additionally, SB-3CT (25 μM) (Selleck, Houston, TX, USA) was added to the complete RPMI-1640 medium at the indicated SW480 cell group for 24 h.

Western blot

SW480 cell and Caco-2 cell lysates were performed using RIPA lysis buffer (Solarbio Science and Technology Co, Ltd, Beijing, China), supplemented with phosphatase and protease inhibitors. Protein concentration was quantified using a BCA protein assay kit. Equivalent amounts of protein were separated by sodium dodecyl sulfate-polyacrylamide gel electrophoresis (SDS-PAGE) and transferred onto polyvinylidene difluoride (PVDF) membranes (Millipore, Burlington, MA, USA). Membranes with blot proteins were incubated overnight at 4 °C with anti-MMP-2 antibody (Servicebio, Wuhan, China), anti-PD-L1 antibody (Servicebio, Wuhan, China) or anti-GAPDH antibody (Servicebio, Wuhan, China) at a dilution of 1:1,000. Secondary antibodies, conjugated to horseradish peroxidase (Cell Signaling Technology, Boston, MA, USA), were used at a dilution of 1:2000. Protein bands were detected using a Chemi-Scope exposure machine. The protein bands were analyzed using the ImageJ software (NIH, Bethesda, MD, USA). The relative expression levels of MMP2 and PD-L1 were quantified by the band densities to that of glyceraldehyde-3-phosphate dehydrogenase (GADPH).

Cell viability assay

To assess the effects of MMP2 downregulation on cell proliferation. SW480 cells with varying treatments were seeded in the 96-well plate. Cell proliferation was measured after 24 h using the Cell Counting Kit-8 (CCK-8) assay kit (Glpbio, Montclair, CA, USA), following the manufacturer’s protocol. The absorbance at 450 nm was measured using a microplate reader.

Cell invasion assay

To further assess the effects of MMP2 downregulation on cell invasiveness, SW480 cells were transfected with MMP2-siRNA or control siRNA. For the invasion assay, cells were subjected to a transwell assay (Corning, NY, USA) with Matrigel-coated inserts. The upper transwell chamber contained medium without FBS, whereas the lower chamber contained medium with 15% FBS. After cultured for 24 h, the number of invading cells was quantified.

Correlation between key genes and immune infiltration scores

Immune infiltration scores in tumor were analyzed using CIBERSORT. The correlation between target gene and expression matrix or immune cells was analyzed using Spearman correlation algorithm, with statistical significance defined as p < 0.05.

Statistical analysis

All statistical analyses were conducted using R software (version 3.4.0.3; https://www.r-project.org/), while significance was determined with p < 0.05. Data are presented as the mean ± standard deviation (SD). DEGs analysis was performed using the top Table and decide Test functions provided by the “limma” package to summarize linear model results, perform hypothesis tests, and adjust p-values to perform multiple tests. Spearman’s rank correlation was used to determine the relationships of immune gene expression and immune infiltration scores, and p < 0.05 was considered statistically significant.

Results

Screening of DEGs between high- and low-CD8+ T cell-infiltrated patients

The immune microenvironment plays a crucial role in the pathogenesis and progression of COAD. Understanding the differences in gene expression between patients with high and low CD8+ T cell infiltration may provide insights into the mechanisms underlying immune evasion and tumor development. In this study, using the CIBERSORT method, the immune landscape of 24 immune cell types in COAD patients was estimated in three independent datasets: TCGA, GSE197802, and GSE140973 (Figs. 1A, S1, S2). To investigate the impact of CD8+ T cell infiltration on gene expression patterns, COAD patients were categorized into high- and low- CD8+ T cell groups based on whether their CD8+ T cell infiltration percentage was above or below the median value, respectively (Fig. 1B). The down-regulated and up-regulated DEGs between the high- and low-CD8+ T cell infiltration groups were visualized in the volcano plots (Figs. 1C–1E). Notably, a set of 134 genes was significantly upregulated in patients with low-CD8+ T cell infiltration (Fig. 1F), suggesting their potential involvement in shaping an immune-suppressive TME or regulating CD8+ T cell exclusion.

Figure 1 Analysis of immune cell infiltration and screening of DEGs between high- and low-CD8+ T cell infiltrated patients.

(A) The abundance ratio of immune cells in the COAD samples in the TCGA dataset. The horizontal axis represents different patients, and the vertical axis represents different proportions of infiltrated immune cells. (B) The COAD patients were divided into two groups (high and low infiltration) based on the level of CD8+ T cell infiltration. The immune enrichment score was calculated for each group across three datasets: TCGA, GSE197802, and GSE140973. (C–E) Volcanic map of DEGs between high and low- CD8+ T cell infiltrated groups. (F) The 134 DEGs correlated with the low CD8+ T cell infiltration were presented with Upset R diagram. **p < 0.01.

Pathway analysis of DEGs between high- and low-CD8+ T cell-infiltrated patients

To understand the biological significance of these DEGs and their potential roles in COAD progression, GO and KEGG pathway enrichment analyses were performed to determine the biological functions of screened DEGs in patients with high- and low-CD8+ T cell infiltration. GO analysis indicated that DEGs were significantly enriched in Jak-STAT signaling pathway, TNF-α signaling pathway and lymphocyte mediated immunity and other related pathways with oncogenic activation and immunosuppression (Fig. 2A). Concurrently, KEGG pathway enrichment analysis demonstrated that these genes were mostly enriched in pathways associated with angiogenesis and carcinogenesis including Rap1 signaling pathway, vascular endothelial growth factor (VEGF) signaling pathway and transcriptional mis-regulation in cancer (Fig. 2B). Additionally, GSEA pathway enrichment analysis was constructed in high- and low-CD8+ T cell infiltrated patients. The results indicated that low CD8+ T cell infiltration was associated with significant downregulation of immune-related pathways, suggesting a suppressed immune response in these patients (Figs. 2C–2E).

Figure 2 Functional enrichment analysis of DEGs associated with low CD8+ T cell infiltration.

(A) GO pathway enrichment analysis of DEGs. The horizontal axis represents the gene ratio of enrichment pathway; the vertical axis represents specific pathways of enrichment; the size of the circle represents the number of enrichments; the color represents p value. (B) KEGG pathway enrichment analysis of DEGs. The horizontal axis represents the gene ratio of enrichment pathway; the vertical axis represents the specific pathway of enrichment; the size of the circle represents the number of enrichments; the color represents p value. (C–E) The gene set enrichment analysis (GSEA) was conducted between high- and low-CD8+ T cell infiltrated groups. The core pathways were selected, p value < 0.05, FDR < 0.25.

Identification of hub genes and relationship between hub genes and OS

Given the potential importance of DEGs in immune regulation and tumor progression, we aimed to explore their relationship with immune checkpoints, which are key molecules in tumor immune evasion. We conducted correlation analysis between the 134 DEGs and CD274 (PD-L1) expression, and finally identified 42 hub genes, which were significant positively correlative with CD274 (Fig. 3A), highlighting their potential involvement in immune evasion mechanisms. To further assess the clinical relevance of these hub genes, we then explored their correlation with the OS of COAD patients. Among the 42 hub genes, results revealed that MMP-2 was significantly correlated with poor OS of COAD patients (HR = 1.79; p = 0.018), suggesting that MMP-2 may serve as a critical prognostic marker for COAD (Figs. 3B, S3A).

Figure 3 Identification of hub genes and relationship between these genes and OS of COAD patients.

(A) The correlation between the low-CD8+ T cell infiltration related genes and CD274 were conducted and presented with single gene co-expression heat map. The X-axis of this graph represents the z-score quantification of gene expression values, and is indicated by different color bar, with blue representing lower and red representing higher. The Y-axis represents the Log value of CD274 gene expression. hub genes: n = 42. *p < 0.05, **p < 0.01, ***p < 0.001. (B) The correlation between the hub genes and the OS of COAD patients. Red line indicates higher gene expression, and blue indicates lower gene expression.

Relationship between MMP-2 expression and clinical prognosis of COAD patients

To further explore the clinical relevance of MMP-2 expression, we divided patients into MMP-2 high and low groups based on the median MMP-2 gene expression value. Our analysis confirmed that MMP-2 was significantly negatively correlative with OS (HR = 1.78, p = 0.02) and progress-free interval (HR = 1.77, p = 0.04) of COAD patients (Figs. 4A, 4B). Additionally, results indicated that high MMP-2 expression was also associated with adverse clinical prognostic features, including perineural invasion, lymphatic invasion, N stage and carcinoembryonic antigen (CEA) level in COAD patients (Figs. 4C–4F). These findings underscore the potential of MMP-2 as a negative prognostic indicator in COAD.

Figure 4 The relationship between MMP-2 expression and survival, clinical characteristics of COAD patients.

(A and B) The Kaplan–Meier analysis of OS and progress free interval comparing the high and low expression groups of MMP-2. p < 0.05, HR > 1. (C–F) The correlation between MMP-2 and clinical characteristics, including perineural invasion, lymphatic invasion, clinical N stage and CEA level. *p < 0.05, **p < 0.01, ns, not significant (p > 0.05).

MMP-2 knockdown inhibited PD-L1 expression in colon cancer cells

Given the observed correlation between MMP-2 and PD-L1 expression, we further determined the relationship between these two molecules. Our results showed that MMP-2 showed a significant positive correlation with CD274 (p < 0.001) (Fig. 5A). To investigate whether MMP-2 inhibition could downregulate PD-L1 in colon cancer cells, we conducted in vitro experiments using the SW480 colon cancer cell line. SW480 colon cancer cell line was transfected with MMP-2 siRNA to knock down MMP-2. The results demonstrated that compared with control siRNA-transfected cells, MMP-2 siRNA-transfected SW480 cells resulted in lower protein levels of MMP-2 (MMP-2 siRNA group vs. control siRNA group, p < 0.01) and PD-L1 (MMP-2 siRNA group vs. control siRNA group, p < 0.05) (Figs. 5B–5D). Furthermore, identical findings were observed using MMP2-siRNA transfection in the Caco-2 cell line and the MMP2 selective inhibitor SB-3CT in SW480 cells (Figs. S4, S6). We further investigated the impacts of MMP2 knockdown on the proliferation and invasion capacity of SW480 cells. The results showed that compared to control siRNA-transfected cells, MMP-2 siRNA-transfected SW480 cells resulted in lower cell viability (MMP-2 siRNA group vs. control siRNA group, p < 0.001) and lower invasion cells per view (MMP-2 siRNA group vs. control siRNA group, p < 0.0001) (Fig. S5). These results suggest that targeting MMP-2 may be a variable strategy to modulate PD-L1 expression, thereby enhancing anti-tumor immune responses in COAD via some form of immune-mediated cell death.

Figure 5 MMP-2 was positively correlated with CD274 and knockdown of MMP-2 downregulated PD-L1 expression in colon cancer cells.

(A) The correlation between MMP-2 expression and CD274 expression. R > 0.3, p < 0.05. (B) SW480 cells were transfected with MMP-2 siRNA, the level of MMP-2 and PD-L1 were detected by Western blot. (C) Relative expression of MMP-2, n = 3/group. (D) Relative expression of PD-L1, n = 3/group. **p < 0.01, ***p < 0.001, ns, not significant (p > 0.05).

Correlation between MMP-2 expression and immune cell infiltration

To further examine the immunological implications of MMP-2, we divided the patients into two groups according to the high and low expression of MMP-2, and then performed immune infiltration analysis. The results indicated that the infiltration proportions of natural killer (NK) cells, dendritic cells (DCs), neutrophils, macrophages were markedly lower in the high MMP-2 expression group (p < 0.01) (Fig. 6A). Importantly, a negative correlation between MMP-2 and CD8+ T cell infiltration was observed (Fig. 6B), reinforcing the potential role of MMP-2 in promoting an immunosuppressive tumor microenvironment by limiting CD8+ T cell infiltration.

Figure 6 The correlation between MMP-2 and various immune cells.

(A) Analysis of immune cell infiltration between high- and low-MMP-2 expression groups in COAD patients. The horizontal axis indicated immune cells; the vertical axis indicated the proportion of immune cells. R > 0.3, p < 0.05. (B) The correlation between MMP-2 expression and CD8+ T cells. ***p < 0.001, ns, not significant (p > 0.05).

Discussion

COAD remains one of the leading causes of cancer-related deaths worldwide, and is characterized by a high degree of malignancy and a complex tumor microenvironment (Arnold et al., 2017; Kuipers et al., 2015). The high heterogeneity of COAD poses significant challenges to the development of effective therapeutic strategies, particularly in advanced stages. Although immune checkpoint blockade (ICB) therapies, such as PD-L1 inhibition, have been identified as effective treatment strategies for various cancers, their efficacy as monotherapies in COAD is limited due to the dynamic and immunosuppressive nature of the tumor microenvironment (Passardi et al., 2017). Therefore, it is imperative to explore novel therapeutic targets that can synergize with PD-L1 inhibitors to enhance treatment efficacy and improve clinical outcomes for COAD patients.

This article underscores the importance of MMP-2 as a promising therapeutic target in COAD, particularly in the context of immunotherapy. Traditionally, MMP-2 is known for its role in promoting tumor invasion, metastasis, and angiogenesis by degrading the extracellular matrix and basement membrane. However, emerging evidence indicates that MMP-2 is also involved in shaping the tumor immune microenvironment, thereby influencing immune evasion mechanisms, including those mediated by PD-L1 (Hu et al., 2023). The present study provides the first evidence of a positive correlation between MMP-2 expression and PD-L1 level in COAD, along with a negative correlation with tumor-infiltrating lymphocytes (TILs). This highlights the dual role of MMP-2 in promoting tumor progression and modulating immune responses, thereby suggesting it as a critical player in COAD pathogenesis, revealing its potential as a novel therapeutic target.

Recent studies have highlighted the critical roles of multiple members of the MMP family in shaping the TME and facilitating immune evasion. In addition to MMP-2, MMP-9 has been extensively reported to promote tumor invasion and metastasis in CRC and other malignancies, while also playing a role in immune regulation. MMP-9 contributes to ECM degradation, the release of pro-inflammatory cytokines such as TGF-β, and subsequent modulation of immune cell infiltration (Kessenbrock, Plaks & Werb, 2010). Furthermore, MMP-3 has been shown to induce epithelial-to-mesenchymal transition (EMT), and enhance the degradation of ECM (Radisky & Radisky, 2010). However, there is no direct evidence that MMP-3 promotes an immunosuppressive microenvironment. Additionally, MMP-14 (also known as MT1-MMP) has been implicated in tumor progression and is closely associated with tumor-associated macrophages (TAMs) in various solid tumors (Niland, Riscanevo & Eble, 2021). These findings suggest that members of the MMP family may exhibit cooperative regulatory effects on the tumor immune microenvironment. However, compared to the well-established mechanisms linking MMP-2 to PD-L1 expression, the specific roles of other MMPs in COAD immune modulation remain less well characterized. Future studies should aim to elucidate whether additional MMPs share regulatory pathways similar to MMP-2 and investigate potential synergistic interactions among MMPs in modulating immune responses. A deeper understanding of these mechanisms may facilitate the development of novel therapeutic strategies to optimize immune checkpoint blockade therapies in CRC.

In the present study, to explore how CD8+ T cell infiltration influences gene expression within the TME, using a comprehensive bioinformatics approach, we analyzed sequencing data of COAD patients to estimate the infiltration levels of 22 immune cell types, and identified DEGs significantly associated with CD8+ T cell infiltration. Our findings revealed that these DEGs were enriched in pathways associated with oncogenic activation and immunosuppression, providing insights into tumor-immune interactions and potential therapeutic targets. By first identifying DEGs based on immune infiltration rather than prognosis, we established a foundational dataset for subsequent prognostic analyses. Notably, among these DEGs, MMP-2 was further identified as being significantly correlated with poor OS and prognosis in COAD patients, thereby highlighting its potential as an immune-related biomarker and a prognostic indicator.

In vitro experiments based on the COAD-derived cancer cell lines SW480 and Caco-2 further confirmed the role of MMP-2 in regulating PD-L1 expression in COAD. Specifically, MMP-2 selective MMP-2 inhibitor SB-3CT or siRNA-mediated knockdown of MMP-2 in colon cancer cell lines resulted in a significant reduction in PD-L1 expression, a crucial immune checkpoint protein, suggesting a regulatory axis between MMP-2 and PD-L1 in COAD. This finding is particularly relevant because PD-L1 expression on tumor cells is known to cause T cell exhaustion by interacting with PD-1, thereby facilitating tumor immune evasion and reducing the efficacy of immune surveillance and immune killing (Juneja et al., 2017; Ribas, 2015; Zhang et al., 2018). In other words, the suppression of PD-L1 expression plays critical roles in anti-tumor progression (Iwai et al., 2002). Several studies provide insights into the potential mechanisms underlying this relationship between MMP-2 and PD-L1. For instance, studies have demonstrated that MMP-2 can activate and release latent TGF-β via proteolytic cleavage of the latency-associated peptide (LAP) and the large latent TGF-β1 binding protein-1 (LTBP1) (Costanza et al., 2017; Dallas et al., 2002; Ge & Greenspan, 2006). Additionally, the TGF-β signaling pathway plays a pivotal role in EMT, which has been implicated in the regulation of PD-L1 expression (Jiang & Zhan, 2020; Xu, Lamouille & Derynck, 2009). Moreover, TGF-β is able to increase the level of PD-L1 expression by inducing EMT, thereby contributing to immune evasion (Evanno et al., 2017; Jiang & Zhan, 2020). Therefore, the observed down-regulation of PD-L1 following MMP-2 inhibition may be attributed to the suppression of TGF-β and subsequent EMT processes. Simultaneously, our study also investigated the impact of MMP-2 downregulation on CRC cell proliferation and invasion. Our results revealed that knockdown of MMP2 in SW480 cells significantly decreased their cell viability and invasive capacity. This finding aligns with previous research indicating a connection between MMP-2 and cell cycle regulation (Son & Moon, 2010; Webb et al., 2017), and the transwell assay results further confirm MMP-2’s role in promoting cancer cell invasion by degrading the ECM (Han et al., 2008; Radisky & Radisky, 2010). This suggests that inhibiting MMP-2 may enhance CRC cells’ sensitivity to immune-mediated cell death, presenting a potential therapeutic strategy.

The type, density and location of immune cells were closely related to tumor progression and the clinical outcome of patients with CRC (Bindea et al., 2013; Galon et al., 2006, 2012a, 2012b; Mlecnik et al., 2010). The negative correlation between MMP-2 expression and TILs, including NK cells, DCs, neutrophils, macrophages, and CD8+ T cells, further emphasizes the role of MMP-2 in modulating the tumor immune microenvironment. NK cells are crucial components of innate immunity and play a vital role in suppressing tumor growth and metastasis by directly killing cancer cells and indirectly influencing innate and adaptive immune responses (Sivori et al., 2021). DCs are essential for activating CD8+ T cells through antigen presentation, thereby inducing cytotoxic T cell responses to inhibit tumor development (Gardner & Ruffell, 2016). Moreover, activated CD8+ T cells have been proven to be essential in mediating anti-tumor cytotoxicity, and their infiltration into the tumor microenvironment is associated with favorable clinical outcomes in various cancers (Golstein & Griffiths, 2018). Neutrophils also can be involved in tumor immunity and promote cancer progression (Powell & Huttenlocher, 2016). Conversely, higher levels of tumor-associated macrophages, particularly M2 phenotype, are linked to poor prognosis in multiple malignant tumors due to their immunosuppressive effect (Komohara et al., 2016). In this study, our results indicated that MMP-2 high expression is negatively correlative with TILs, which may hinder the infiltration of these TILs into the tumor microenvironment, thereby driving immune escape and tumor progression.

The mechanistic link between MMP-2 and TGF-β activation provides a plausible explanation for the observed negative correlation between MMP-2 and TILs. As mentioned above, MMP-2 is considered as a protein activating TGF-β in tumor. Additionally, activated TGF-β signaling pathway in peritumoral fibroblast can lead to the deposition of collagen fibers that wrap around the tumor, and thus create a physical barrier that hampers T cell infiltration in tumor (Mariathasan et al., 2018). Therefore, we hypothesized that MMP-2 inhibition could recruit TILs into COAD by suppressing TGF-β activation and reducing ECM remodeling to inhibit tumor progression. This hypothesis aligns with recent studies demonstrating that modulating the tumor stroma can significantly impact immune cell infiltration and improve the efficacy of immunotherapy (Chakravarthy et al., 2018; Mariathasan et al., 2018; Salmon et al., 2012; Tauriello et al., 2018).

Taken together, our findings provide compelling evidence that MMP-2 inhibition therapy may enhance the efficacy of anti-PD-L1-based immunotherapy by simultaneously reducing PD-L1 expression and improving TILs infiltration in COAD. The combination of MMP-2 inhibitors with PD-L1 blockade could represent a novel therapeutic strategy for overcoming resistance to monotherapy and enhancing anti-tumor immunity in COAD. Notably, the PD-L1+TIL− tumor tends to develop resistance to monotherapy of anti-PD-L1, as the absence of TILs limits the effectiveness of ICB, while PD-L1 blockade in combination with other therapy increasing T cell infiltration in tumor bed would reduce resistance (Teng et al., 2015). In such cases, combining MMP-2 inhibition with PD-L1 blockade may help to recruit more TILs into the tumor bed, thereby remodeling the immune microenvironment and reducing resistance to immunotherapy.

Several studies have highlighted the potential of combining MMPIs and ICIs in cancer treatment. For example, combination treatment of a MMPI and anti-CTLA-4 antibody has been shown to inhibit tumor growth and metastases in breast cancer model in mice (Li et al., 2016). Moreover, compared to monotherapy of anti-CTLA-4 antibody, this combination treatment increased CD8+/CD4+ ratio in T cells and reduced the number of regulatory T cells and myeloid-derived suppressor cells within the tumor microenvironment, enhancing anti-tumor immunity (Li et al., 2016). Similarly, Ye et al. (2020) demonstrated that a small-molecule MMP-2/MMP-9 inhibitor SB-3CT improved the therapeutic effect of PD-1 or CTLA-4 inhibition in melanoma and lung cancer model. We obtained similar results by using SB-3CT in the SW480 cell line in our study. These findings support the rationale for exploring combination therapies targeting both MMP-2 and PD-L1 in COAD to achieve synergistic anti-tumor effects.

However, there are several limitations to our study that warrant further researches. First, while our in vitro experiments provide evidence for the regulatory role of MMP-2 in PD-L1 expression, these findings need to be validated in vivo to ensure their relevance in the complex tumor microenvironment of COAD. Second, the exact molecular mechanisms through which MMP-2 modulates PD-L1 expression and TILs infiltration remain to be thoroughly elucidated. Future research should focus on dissecting the downstream signaling pathways involved in MMP-2-mediated immunosuppression to identify additional therapeutic targets. Third, our study primarily utilized publicly accessible datasets and cell line models, which may not completely depict the heterogeneity and complexity of COAD in patients. While the GSEA results suggest a significant link between low CD8+ T cell infiltration and immune pathway suppression, these findings cannot be exclusively attributed to CD8+ T cell levels. Other confounding factors may also influence immune suppression, and GSEA alone cannot encompass the full complexity of immune responses. To address potential biases, we plan to conduct multivariate analyses to account for these confounders and validate the role of reduced CD8+ T cell infiltration in immune activation. However, further functional validation is needed to confirm these findings. Therefore, clinical studies involving patient-derived samples and comprehensive immunophenotyping are required to corroborate our findings and assess the therapeutic potential of targeting MMP-2 in combination with PD-L1 inhibitors.

Despite these limitations, our study provides a novel insight into the potential of MMP-2 as a therapeutic target to potentiate the efficacy of anti-PD-L1 treatment in COAD. The findings presented here not only highlight the necessity of understanding the interaction between tumor-associated proteases and immune checkpoint molecules but also open new avenues for the development of combination immunotherapies capable of efficiently overcoming COAD resistance. Future research ought to encompass preclinical and clinical studies to assess the safety, efficacy, and optimal combination regimens of MMP-2 inhibitors and ICIs. Additionally, identifying biomarkers that predict responsiveness to such combination therapies will be critical for patient stratification and personalized treatment approaches in COAD.

Conclusion

In summary, our study demonstrates that high MMP-2 expression is linked with poor prognosis in COAD and correlates with PD-L1 expression and TILs infiltration. Targeting MMP-2 may serve as a promising treatment strategy to potentiate the efficacy of anti-PD-L1 therapy by modulating the tumor immune microenvironment in COAD. Our study provides compelling evidence for further investigating MMP-2 as a therapeutic target in COAD, as well as underscoring the potential of combination immunotherapies to improve treatment outcomes in cancer patients.

Supplemental Information

Supplemental Information 1 The raw data of western blot in Figure 5C & 5D.

The western blot raw data of relative expression of MMP-2/GAPDH and PD-L1/GAPDH.

Supplemental Information 2 Uncropped blots of MMP2, PD-L1 and GAPDH protein.

Supplemental Information 3 Go analysis data.

Supplemental Information 4 GSEA enrichment analysis.

Supplemental Information 5 R code.

Supplemental Information 6 KEGG analysis data.

Supplemental Information 7 DEGs.

Supplemental Information 8 Uncropped blots of Figures S4 and S6.

Uncropped blots of MMP2, PD-L1, and GAPDH protein.

Supplemental Information 9 The abundance ratio of immune cells in the COAD samples in GSE197802.

The horizontal axis represents different patients, and the vertical axis represents different proportions of infiltrating immune cells.

Supplemental Information 10 The abundance ratio of immune cells in the COAD samples in GSE140973.

The horizontal axis represents different patients, and the vertical axis represents different proportions of infiltrating immune cells.

Supplemental Information 11 Additional correlation analysis between hub genes and OS of COAD patients.

The remaining hub genes from Figure 3B are presented here. The red line indicates higher gene expression, and the blue line indicates lower gene expression.

Supplemental Information 12 MMP2 knockdown reduces PD-L1 expression in Caco-2 cell line.

(A) Representative Western blot images showing MMP2 and PD-L1 protein levels in Caco-2 cells transfected with Control siRNA or MMP2-targeting siRNA (MMP2-siRNA). (B) Quantification of MMP2 expression relative to GAPDH in Control siRNA and MMP2-siRNA groups, n = 3/group. (C) Quantification of PD-L1 expression relative to GAPDH in Control siRNA and MMP2-siRNA groups, n = 3/group. **p < 0.01, ***p < 0.001, compared to the Control siRNA group.

Supplemental Information 13 The effect of MMP2 downregulation on cell proliferation and invasion in SW480 cells.

(A) CCK-8 assay showing cell proliferation in Non-transfected, Control siRNA, and MMP2-siRNA transfected SW480 cells. (B) Transwell invasion assay showing the invasive capacity of Non-transfected, Control siRNA, and MMP2-siRNA transfected SW480 cells, magnification of ×200. (C) Average invasive cell number per field. ***p<0.001, ****p<0.0001, ns, not significant (p>0.05), compared to Control siRNA.

Supplemental Information 14 Downregulation of MMP2 by SB-3CT reduces PD-L1 expression in SW480 cell line.

(A) SW480 cells were treated with MMP-2 selective inhibitor SB-3CT (25 μM, MMP2-inh group), the level of MMP-2 and PD-L1 were detected by Western blot. (B) Relative protein expression of MMP-2, n=3/group. (C) Relative protein expression of PD-L1, n=3/group. **p < 0.01, ns, not significant (p>0.05), compared to Control siRNA.

Additional Information and Declarations

Competing Interests

The authors declare that they have no competing interests.

Author Contributions

Yiyi Xiao conceived and designed the experiments, performed the experiments, analyzed the data, prepared figures and/or tables, authored or reviewed drafts of the article, and approved the final draft.

Guangming Li conceived and designed the experiments, performed the experiments, analyzed the data, prepared figures and/or tables, authored or reviewed drafts of the article, and approved the final draft.

Yongjie Xie performed the experiments, analyzed the data, prepared figures and/or tables, and approved the final draft.

Bo Shao performed the experiments, prepared figures and/or tables, and approved the final draft.

Jingpeng Hao performed the experiments, prepared figures and/or tables, and approved the final draft.

Yanglin Zhu performed the experiments, prepared figures and/or tables, and approved the final draft.

Dejun Kong performed the experiments, prepared figures and/or tables, and approved the final draft.

Yafei Qin performed the experiments, prepared figures and/or tables, and approved the final draft.

Hong Qin performed the experiments, prepared figures and/or tables, and approved the final draft.

Shaohua Ren performed the experiments, prepared figures and/or tables, and approved the final draft.

Hongda Wang performed the experiments, prepared figures and/or tables, and approved the final draft.

Chenglu Sun performed the experiments, prepared figures and/or tables, and approved the final draft.

Hao Wang conceived and designed the experiments, performed the experiments, analyzed the data, prepared figures and/or tables, authored or reviewed drafts of the article, and approved the final draft.

Data Availability

The following information was supplied regarding data availability:

The data is available at TCGA and NCBI GEO: GSE197802, GSE140973 and raw measurements are available in the Supplemental Files.

The raw data is available at figshare: Xiao, Yi-Yi; Li, Guang-ming; Xie, Yong-jie; Shao, Bo; Hao, Jing-peng; Zhu, Yang-lin; et al. (2024). High matrix metalloproteinase-2 expression predicts poor prognosis of colon adenocarcinoma and is associated with PD-L1 expression and lymphocyte infiltration. figshare. Figure. https://doi.org/10.6084/m9.figshare.27324075.v5.

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
