# Peer review of "High matrix metalloproteinase-2 expression predicts poor prognosis of colon adenocarcinoma and is associated with PD-L1 expression and lymphocyte infiltration"

_PeerJ, doi:10.7717/peerj.19550_

## Round 0.1 · original submission · Major Revisions

Please address the concerns of all reviewers and amend manuscript accordingly

Reviewer 1 ·

Basic reporting

The study is generally clear and well-structured and the background information and references provided in this article are comprehensive and complete.

Experimental design

The experimental design presented in this article is largely sound and offers a relatively comprehensive description of the experimental methods.For Western blot analyses, the original, uncropped gel images should be provided.

Validity of the findings

This study presents MMP-2's role in the immune microenvironment of COAD and its potential as a therapeutic target to enhance anti-PD-L1 immunotherapy. There have already been numerous studies reporting the relationship between MMP-2 and immunotherapy.For example, the article titled "Small-molecule MMP2/MMP9 inhibitor SB-3CT modulates tumor immune surveillance by regulating PD-L1" has already reported that inhibiting MMP2/MMP9 can influence PD-L1 expression. The majority of this manuscript relies on bioinformatics analyses derived from public databases, with minimal experimental validation of the conclusions. The authors should substantiate their findings by conducting additional experiments using animal models and patient specimens.

Additional comments

NO

Reviewer 2 ·

Basic reporting

In this manuscript the authors found that MMP-2 not only contributes to tumor progression but also plays a critical role in the immunosuppressive microenvironment of colon adenocarcinoma (COAD). The demonstrated association between MMP-2 and PD-L1 expression, along with its effect on tumor-infiltrating lymphocytes (TILs), indicates that MMP-2 is a promising alternative target for improving the efficacy of anti-PD-L1 immunotherapy. Targeting MMP-2 may over a novel avenue for overcoming resistance to conventional immunotherapies in COAD patients. The study's experimental design is reasonable. These are very impressive findings that support the conclusion and will be of interest to the readers of the journal. This study has a certain-degree of originality, and the rationality of this study is well-presented. Meanwhile, the manuscript is well-written and the experiments are clearly presented. However, there are minor revisions should be provided to further improve the manuscript.
- The authors demonstrated that MMP-2 was involved in shaping the tumor immune microenvironment, thereby influencing immune evasion mechanisms. There are multiple members of the MMPs family of proteins, whether other members of MMPs have similar regulatory roles, such as MMP-3, -9, -11, and etc. Please provide further elucidation in the Discussion section.
-In the Materials and Methods section, lines 219-229, please describe the quantitative analysis of MMP2 and PDL1 by Western blot in more detail.
-Please provide the gene sequence of the MMP-2 siRNA in the Materials and Methods section.
-In Figure 3A and 5C, please indicate the number of samples in the Figure legends.
-In Figure 5B, please add the molecular weight of target proteins or markers.

Experimental design

The study's experimental design is reasonable.

Validity of the findings

These are very impressive findings that support the conclusion.

Additional comments

This study has a certain-degree of originality, and the rationality of this study is well-presented.

Reviewer 3 ·

Basic reporting

The manuscript is clearly presented overall and is well suited for the journal. The introduction and discussion sections were thorough and easy to follow. I also appreciate that the authors have also discussed the limitations of their study.

1. In “Conclusion” section, line 78, “perhaps” is not an often-seen word in academic writing and it would be better to replace it with “potentially” or “subsequently”.
2. Line 116: there is an additional “q” in front of “quite”.
3. Line 122: there should be an “and” between “lymphatic invasion” and “lymph node metastasis”.
4. Line 146: “Given this, MMP-2 is … in cancers”. I suggest removing this sentence, as this paragraph is focused on CD8+ T cells and it doesn’t extensively discuss the relationship between MMP-2 and CD8+ T cells. It feels abrupt to bring up MMP-2 at the end of this paragraph. The motivation of studying MMP-2 has been mentioned at the end of the previous paragraph (line 133-134).
5. Line 173 please add citation to CIBERSORT algorithm.
6. Line 197 and 199: “Selected the KEGG Pathways…” and “Selecting c2.cp.kegg…”, these are not proper structure of English sentences. Please correct the grammar.
7. Line 300: “negatively” should be “negative”.
8. Line 307: I recommend replacing “severe” with “high”.
9. Line 309-310: “Despite ICB… have been identified as effective…” The grammar is not correct. It should be “Despite that ICB… have been identified …”.
10. Figure 1A - legend font is too small and not readable.
11. Figure 1 panel C-E - it is advisable to use the same color as CD8_L among the 3 panels, and the same color as CD8_H. Now there are three colors for either category.
12. Figure 3 fonts are too small to read. There are too many panels in B and all of them are too small to read. The authors may consider only retaining a few panels of B and save the others into supplementary figure.
13. Figure 3A: it is not clear to me how CD274 low and high is quantified and defined. It is not clear what the x axis represents.

Experimental design

I appreciate that the authors have provided original images of Western Blot and have repeated the experiment 3 times.

1. The in vitro validation experiment was only conducted using one siRNA of MMP-2 and performed in only one cell line. Please perform MMP-2 knockdown using at least one additional siRNA sequence. Depending on the authors’ timeline, they should also consider performing the experiment in another colon cancer cell line.
2. In Section “Cell culture and transfection”, please include the siRNA sequences of control siRNA and siMMP-2. If they are proprietary products from a company and that their sequences are not available, please indicate that in the manuscript.

Validity of the findings

1. Line 248: please clarify how are COAD patients categorized into high vs low CD8+ T cell groups. Is it quantified by the % of CD8+ T cell in the overall immune cell composition? What is the cutoff and how is it determined? I did not find very clear explanation in the main text or method section.
2. Line 279: Figure 4A and 4B seem to have divided patients into the MMP-2-high and MMP-2-low group, however I did not find the criteria of defining an MMP-2 cutoff for such classification. Please explain.
3. Line 194 there are two p value cutoffs, please clarify.

Reviewer 4 ·

Basic reporting

1. Line 116: spell check: “qquite” inconsistent. I would rather just say inconsistent and give a reference here.
2. The authors should discuss the potential ways by which regulatory role of MMP-2 regulates PD-L1 expression in the discussion section.
3. Even though the authors used professional english throughout, the authors need to be a little more cautious with explaining why are they doing the experiments they are, and what does it show.

Experimental design

Specific comments:
1. Explain Fig 1B in the Figure legend. Just saying The COAD patients were divided into two groups, respectively, according to the level of CD8+ T cell infiltration needs to be rephrased. The authors should clarify the 2 datasets GSE197802 and GSE140973 either in the text or in figure legend.
2. The authors need to clearly state why was the screening of DEGs between high and low CD8T cell infiltrated patients needed here, and what does it show?
a. Why was patient prognosis or clinical outcomes not used in the screening of DEGs?
3. Fig2C-E: How did the authors conclude that low CD8+ T cell infiltration could significantly suppress the activation of immune pathways? Just a gene set enrichment analysis (GSEA) between high- and low-CD8+ T cell infiltrated groups does not tell you that. Wouldn’t there be a bias between the 2 groups which are basically selected based on CD8T cell infiltration?
4. Fig4: Could the authors confirm the increased protein expression of MMP2 in patient samples (vs normal colorectal tissues)? Or give a reference showing that the protein expression of MMP2 is increased in COAD patients?
5. Fig 5B: Does overexpression of MMP2 increases PDL2 expression as well?
a. Could you show PDL-1 downregulation using MMP2 inhibitors in the SW480 cell lines?
b. Could you prove MMP2 downregulation leads to reduction in PDL1 expression in other colon adenocarcinoma cell line like Caco-2 as well?
6. Fig 5: Does MMP2 downregulation inhibit cell proliferation or regulate invasiveness? Basic in vitro experiments could be done here to answer this.
7. Fig 6B: Could you also explain the “purity” correlation to MMP2 expression.

Validity of the findings

All data seem to have been provided.

Additional comments

In this manuscript, Xiao et al investigates the role of MMP2 in Colon Adenocarcinoma (COAD) and identifies it as a critical player in COAD pathogenesis. They highlight the dual role of MMP-2 in promoting tumor progression and modulating immune responses and suggest MMP2’s potential as novel therapeutic target.
The authors go a commendable job using bioinformatics to analyze sequencing data of COAD patients, and thereby identifying MMP2 as a key player associated with oncogenic activation and immunosuppression. The authors seem to be the first ones to demonstrate that high MMP-2 expression is linked with poor prognosis in COAD and correlates with PD-L1 expression and CD8+ T cell infiltration.

Overall, I like that the authors lay the basis for investigating the role of MMP2 in COAD, and even though confirmatory and subsequent experiments with regards to their claims need to be performed- such as experiments related to tumor infiltration, in vivo data, and mechanism (that could be done in a subsequent paper)- I do recommend accepting the manuscript but after addressing my comments.

---

## Round 0.2 · accepted · Accept

All issues pointed out by the reviewers were addressed and the revised manuscript is acceptable now.

Reviewer 1 ·

Basic reporting

The study’s overall structure is logical and well-organized, and the background details and cited references are adequately thorough.

Experimental design

The experimental design is now more complete and meets the required standards.

Validity of the findings

The revised manuscript now provides stronger evidence

Additional comments

NO

Reviewer 3 ·

Basic reporting

The academic English quality is significantly improved in the revised manuscript. The authors have added citations according to my suggestions. The figures are also more easily readable.

Experimental design

The methods are clearly stated and results are clearly presented. The authors further addressed my concerns by repeating the siRNA knockdown experiment of MMP2 in another colon cancer cell line and validated their finding.

Validity of the findings

The authors have provided sufficient supporting data.